**External and Internal Cloud Condensation Nuclei (CCN) Mixtures: Controlled Laboratory**
**Studies of Varying Mixing States.**
Diep Vu[1,2,§], Shaokai Gao[1,♦], Tyler Berte[1,2], Mary Kacarab[1,2], Qi Yao[4], Kambiz Vafai[3] and Akua Asa-Awuku[1,2,4,*]
1.*Department of Chemical and Environmental Engineering, Bourns College of Engineering, University of
California, Riverside, CA 92521, USA
2.*Bourns College of Engineering, Center for Environmental Research and Technology (CE-CERT), Riverside, CA
92507, USA
3. Department of Mechanical Engineering, Bourns College of Engineering, University of California, Riverside, CA
92521, USA
4.*Department of Chemical and Biomolecular Engineering, A. James Clark School of Engineering, University of
Maryland, College Park, MD 20742.
§. Currently at Ford Motor Company, Research & Innovation Center Dearborn, MI 48124, USA
♦. Currently at Phillip 66 Research Center, Research and Development, Bartlesville, OK 74004, USA
*Correspondence to:* A. Asa-Awuku (asaawuku@umd.edu)
**Abstract**
Changes in aerosol chemical mixtures modify cloud condensation nuclei (CCN) activity. Previous studies have
developed CCN models and validated changes in external and internal mixing state with ambient field data. Here,
we develop an experimental method to test and validate the CCN activation of known aerosol chemical composition
with multicomponent mixtures and varying mixing states. CCN activation curves consisting of one or more
activation points is presented. Specifically, simplified two component systems of varying hygroscopicity were
generated under internal, external, and transitional mixing conditions. $\kappa$-Köhler theory predictions were calculated

for different organic and inorganic mixtures and compared to experimentally derived kappa values and respective mixing states. This work employs novel experimental methods to provide information on the shifts in CCN activation data due to external to internal particle mixing from controlled laboratory sources. Results show that activation curves consisting of single and double activation points are consistent with internal and external mixtures, respectively. In addition, the height of the plateau at the activation points are reflective of the externally mixed concentration in the mixture. The presence of a plateau indicates that CCN activation curves consisting of multiple inflection points are externally mixed aerosols of varying water-uptake properties. The plateau disappears when mixing is promoted in the flow tube. At the end of the flow tube experiment, the aerosol are internally mixed and the CCN activated fraction data can be fit with a single sigmoidal curve. The technique to mimic external to internally mixed aerosol is applied to non-hygroscopic carbonaceous aerosol with organic and inorganic components. To our knowledge, this work is the first to show controlled CCN activation of mixed non-hygroscopic soot with hygroscopic material as the aerosol population transitions from external to internally mixed states in laboratory conditions. Results confirm that CCN activation analysis methods used here and in ambient data sets are robust and may be used to infer the mixing state of complex aerosol compositions of unknown origin.

## 1. Introduction

Atmospheric cloud condensation nuclei (CCN) are comprised of complex mixtures of organic and inorganic compounds. The chemical and physical diversity present in complex mixtures can significantly complicate the quantification of aerosol-cloud interactions, thereby making it difficult to predict CCN activity (e.g., Riemer et al., 2019). In this work, we define mixtures and the aerosol mixing state as the chemical diversity across an aerosol distribution. Knowledge of the mixing state and the chemical composition can greatly improve CCN predictions and has been the focus of several studies (e.g., but not limited to (Bilde and Svenningsson 2004; Abbatt et al. 2005; Henning et al. 2005; Svenningsson et al. 2006; King et al. 2007; Cubison et al. 2008; Kuwata and Kondo 2008; Zaveri et al. 2010; Su et al. 2010; Wang et al. 2010; Spracklen et al. 2011; Ervens et al. 2010; Asa-Awuku et al. 2011; Lance et al. 2012; Liu et al. 2013; Jurányi et al. 2013; Paramonov et al. 2013; Padró et al. 2012; Moore et al. 2012; Meng et al. 2014; Bhattu and Tripathi 2015; Almeida et al. 2014; Schill et al. 2015; Crosbie et al. 2015; Che

et al. 2016; Ching et al. 2016; Mallet et al. 2017; Sánchez Gácita et al. 2017; Cai et al. 2018; Schmale et al. 2018;
Mahish et al. 2018; Kim et al. 2018; Chen et al. 2019; Stevens and Dastoor 2019)
It is well accepted that the water content and the point of activation is dependent on more factors than just the
supersaturation and dry diameter for CCN active aerosols (Dusek et al., 2006; Petters and Kreidenweis, 2007). The
droplet growth and activation of slightly soluble organics can be modified when internally mixed with inorganic
salts that readily deliquesce (Cruz et al., 1998; Padró et al., 2002; Svenningsson et al., 2006). Although inorganic
salts are well characterized, the quantification of CCN activity is complicated when they are internally mixed with a
complex organic. Consequently, simple mixing rules may no longer be appropriate. It has been observed that mixed
aerosols can activate at lower supersaturations than their bulk constituents and organic compounds that may not
traditionally be considered as water soluble may aid in the formation of a cloud droplet by acting as a surfactant,
depressing surface tension, or simply by contributing mass (Cruz et al., 1998; Padró et al., 2007; Svenningsson et al.,
2006). In addition, when there is a sufficiently large enough fraction of salt, the slightly soluble core can dissolve
before activation, thus lowering the required supersaturation (Sullivan et al., 2009). Thus, the mixing state and
extent of mixing can substantially influence CCN activity.
To help minimize the complexity in characterizing aerosol hygroscopic and CCN activation properties, CCN
data analysis has traditionally been simplified by assuming that *i)* the aerosols share a similar or uniform
hygroscopicity over a particle size distribution, *ii)* the CCN particle size can be described by the electrical mobility
diameter, *iii)* CCN consists of few multiply charged aerosols and *iv)* all CCN active aerosols readily dissolve at
activation. As a result, a singular sigmoidal fit is commonly applied over the entire CCN activation. However, this
method of analysis may not be fully representative of the heterogeneous mixing state occasionally present in the
aerosol sample. Thus a CCN mixture refers to the diversity of activated aerosols in the particle population (not the
property of an individual particle; i.e., Winkler, 1973; Riemer et al., 2019).
Previous studies have addressed aerosols with singular or internally mixed binary chemical species (e.g. but not
limited to, Bilde et al., 2004; Broekhuizen et al., 2004; Petters and Kreidenweis, 2007; Shulman et al., 1996;
Sullivan et al., 2009). However, ambient measurements indicate complex aerosol populations consisting of both
external and internal mixtures (e.g., but not limited to Ervens et al., 2007; Lance et al., 2012; Moore et al., 2012,
Padró et al., 2012). By accounting for the mixing states and extent of mixing in field data sets, CCN concentration
predictions can be greatly improved (e.g but not limited to Padró et al., 2012; Wex et al., 2010, Su et a., 2010,
Kuwata and Kondo, 2008). However, dynamic changes in particle mixing states have not been reproduced in the
laboratory and subsequent treatment of CCN measurement and analysis have not been readily studied in depth under
controlled laboratory conditions.

In this work, we seek to improve the experimental CCN activation analysis techniques of complex mixtures by

investigating the influence of mixing state on activation curves with known aerosol composition. Theoretical
postulations have already been developed and applied to ambient data sets (e.g., but not limited to, Su et a., 2010,
Padró et al., 2012;  Bhattu et al., 2016;) but never before has a systematic laboratory experiment controlled and
validated the extent of particle heterogeneity for CCN activation.  To understand the impact of mixing state on CCN
activation, simplified two component mixtures of known compounds with varying hygroscopicities are created
under internal, external, and in-between (transition) mixing conditions. In addition, black carbon containing particles
(BC) and BC mixing state data is presented.  BC is renown for its direct radiative effects yet little is known
experimentally about the contributions of BC mixtures to aerosol-cloud interactions at varied mixing states. Previous
work investigating the contribution of BC to aerosol-cloud interactions at various mixing states has been studied
(e.g., but not limited to Bond et al., 2013 and references therein, Lammel and Novakov 1995; Novakov and Corrigan
1996; Weingartner et al. 1997; Dusek et al. 2006; Kuwata and Kondo 2008; Koehler et al. 2009; McMeeking et al.
2011; Liu et al. 2013; Rojas et al. 2015).  However, many of these have been limited more to ambient related field
studies and less under controlled laboratory settings. Here, the CCN activation and hygroscopic properties of soot
mixed with atmospherically relevant constituents of varying hygroscopicity are investigated under laboratory
controlled conditions.  This work provides evidence on the differences between inorganic, slightly soluble, and
insoluble externally and internally mixed compositions for the uptake of water, and subsequent CCN activity.

## 100    2.   Experimental Methods

### 101    2.1 Aerosol Composition and Sources

The CCN activity of two component aerosol mixtures under internal, external, and combinations of mixing

states are explored in this study. The components include very hygroscopic (inorganics), hygroscopic (organic
acids), and non-hygroscopic (Black Carbon, BC). For known chemical compositions, a Collison-type atomizer
generated singular-component solutions of ammonium sulfate, ($(NH_4)_2SO_4$, Acros 99.5%), sodium chloride, (NaCl,
Acros 99+%) and succinic acid ($C_4H_6O_4$, Acros 99%) and subsequent internal mixture combinations as described in
the sections Aerosol Mixing States and Modified Mixing States: External to Internal. Succinic acid, classified as a
slightly soluble dicarboxylic acid, (18) $(NH_4)_2SO_4$, and NaCl are all relevant model atmospheric compounds with
varying degrees of solubility and hygroscopicity. All atomized solutions were prepared using Millipore[©] DI water
(18 mΩ, TOC ≤ 5 ppb). Atomized wet droplets were dried with silica gel diffusion dryer (as commonly practiced).
In addition, we employ a heated column before the diffusion dryer to facilitate the evaporation of water from the wet
particles. The implications of the use or absence of a heated column are discussed in the results below.
An AVL Particle Generator (APG), which houses a mini Combustion Aerosol STandard soot generator
(miniCAST 6203C, Jing ltd.), was used to generate carbonaceous aerosols. The APG consists of a propane burner
followed by a volatile particle remover. The burner was operated at 400°C with a propane and air flow rate of 15 ml
per min and 1.0 l per min, respectively. The miniCAST utilized in the APG has been well characterized in previous
work (e.g., Pinho et al. 2008; Seong and Boehman 2012; Mamakos et al. 2013; Maricq and Matti Maricq 2014;
Durdina et al. 2016; Moore et al. 2014). The soot formed is a mixture of black and oxidized carbon (Moore et al.
2014). The aerosol structures generated by the APG likely consisted of fractal-like agglomerates of non-spherical
particles (Moore et al. 2014; Durdina et al., 2016). APG Combustion aerosols are mixed with inorganic and slightly
soluble succinic acid and CCN activity is subsequently measured at different supersaturations.
**2.2 Aerosol Mixing Methods**
Mixing compounds in solution can readily form internal mixtures of aerosol (Gibson et al., 2007; Hameri et
al., 2002). Solution mixtures of a highly hygroscopic compound, NaCl, and a slightly hygroscopic compound,
succinic acid, are shown. Five aqueous solutions of succinic acid and NaCl with molar ratios of 100:0, 87:13, 69:31,
43:57, 0:100 were aerosolized using a single atomizer, passed through a heated column, dried, and sampled directly
into a Scanning Mobility Particle Sizer (SMPS) and CCN Counter (CCNC). Instrument specifications are discussed
in section 2.3.
External mixtures were formed via two methods**.** The first and simplest method required two sufficiently dry
aerosol streams to mix. Two aerosol streams were joined via a Swagelok® Tee connector. External mixtures were
also formed in a flow tube mixing apparatus. As conditions (e.g., but not limited to, residence time, temperature,
pressure, relative humidity) change in a flow tube, it is assumed that the external mixture may transition into an
internally mixed aerosol system. A flow tube mixing apparatus was constructed to test this assumption and modify
the extent of mixing of multiple components (Fig. 1 & 2).


**Figure 1.** External and Internal Mixtures with gradual mixing in flow tube


This work shows changes in CCN experimental data and analysis as a result of changes in the extent of

mixing. Results of the CCN activation are presented in the Section: Modified Mixing: External to Internal Mixing in
the Aerosol Flow Tube. A brief description of the flow tube is provided here. The first aerosol stream is introduced
into the flow tube by a ¼ inch stainless steel (SS) tube. The second aerosol stream is also introduced by a ¼ inch SS
tube, but is expanded to an outer concentric ¾ inch SS tube using a SS Swagelok tee connection. The two aerosol
flows are initially mixed together at the exit of ¼ inch tube and aerosol mixes within the ¾ inch SS tube for an
additional 12 inches before entering the quartz tube where it continues to mix. In this study, the pressure and
temperature of the flow tube is maintained at ambient conditions. The extent of mixing in the flow tube mixer has
been modeled by Computational Fluid Dynamics simulation (CFD - Comsol) to test and improve the aerosol mixing
capabilities of the flow tube mixer (Fig. A1). The focus of this work is not the mixing apparatus but the CCN
behavior that results from changes in the extent of mixing.  It is noted that particle losses likely occur within the
flow tube system but do not affect the intrinsic aerosol and CCN properties (activated fractions) presented here.

**Figure 2.** Example of charge corrected a) a single component activation curve b) a multiple activation curve from an

externally mixed/heterogeneous system. The assymptote, η, varies in height and length with the presence of mixed

components and their respective hygroscopicities.

**2.3 CCN Activity and Chemical Composition: Measurements and Instrumentation**

CCN activation is measured with particle sizing and counting instrumentation in parallel with CCN counting

instrumentation. This technique is widespread and has been used in numerous publications (e.g., but not limited to,
Moore et al., 2010; Padró et al., 2012). The development of a single continuous-flow thermal gradient diffusion
column CCN Counter, CCNC (Droplet Measurement Technologies, Inc.) has provided rapid (~1 Hz) and robust
CCN data (Lance et al., 2012; Roberts and Nenes, 2005). Aerosols with a $S_c$ lower than the supersaturation in the
column activate and form droplets. These droplets are detected and counted using an optical particle counter at the
exit of the column.

A TSI 3080 Electrostatic Classifier selects and measures aerosol size distributions.    Polydisperse aerosol

streams are passed through a bi-polar krypton-85 charger and then through a differential mobility analyzer (DMA),
where the aerosols are sized according to electrical mobility with a sheath to aerosol flow ratio of 10:1. The
monodispersed flow is then split to a CPC and a CCN counter. CN concentrations were measured with a
condensation particle counter, CPC (TSI 3772, TSI 3776).

The CCNC is operated at 0.5 lpm with a sheath to aerosol flow ratio of 10:1 and is calibrated with $(NH_4)_2SO_4$ to

determine the instrument supersaturation (Rose et al., 2008). Operating the CCN in parallel with the CPC allows for
the simultaneous measurements of the total CN and CCN of the monodispersed aerosols. By operating the DMA in
scanning voltage mode and maintaining a constant column supersaturation, the CCN/CN, or activation ratio, as a
function of dry diameter can be obtained for a given supersaturation. These size resolved CCN distributions obtained
through scanning mobility CCN analysis (SMCA) are produce CCN activation curves, CCN/CN ratio as a function
of particle mobility diameter (Moore et al., 2010). SMCA produces high resolution CCN activation data near the
50% efficiency critical diameter every ~ 2 minutes.

An Aerodyne high-resolution time-of-flight aerosol mass spectrometer (HR-ToF-AMS) measured the non-

refractory bulk composition (DeCarlo et al., 2006). The HR-ToF-AMS was operated in V-mode to track the
concentration and vacuum aerodynamic diameter as the aerosol fractions were modified.
**3.  CCN Analytical Method**

CCN data analysis of single component aerosols, such as AS, are well characterized. The activation of a single

known component yields a simple sigmoidal activation curve and is often used for instrument calibration (Fig. 2a).
However ambient aerosols generally exist as complex mixtures of organic and inorganic species. CCN data sets
from ambient and chamber studies, which consist of these aerosol mixtures, may not show a single sigmoidal
activation curve but instead can exhibit multiple activation curves not associated with doubly charged particles (Fig.
2b).

Sigmoidal fits are applied to the CCN/CN as a function of dry activation diameters for the multicomponent

aerosols. Externally mixed aerosol fractions in activation curves have been previously observed in ambient studies
by Lance et al. (2013), Moore et al. (2012), and Bougiatioti et al. (2011). For those studies, *E* was defined as the
hygroscopic fraction, and *1-E* the non-hygroscopic mixed fraction. For this study the first curve is similarly defined
as the hygroscopic externally mixed fraction (EMF) with an asymptote, or plateau of $\eta$. The dependence of $\eta$ varies
with the presence of mixed components and their respective hygroscopicities. Thus we evaluate $\eta$ for controlled
compositions and compare how representative they are of the known fractions of mixtures.

A sigmoidal fit through the EMF determines the particle dry diameter of the more hygroscopic species. A

second sigmoidal fit is applied to the second activation curve. An example is shown in Fig. 2b for an external
mixture of AS and SA. A sigmoid is fit for the more hygroscopic species, AS, and then a second for the less
hygroscopic species, SA. The activation diameters are consistent with those expected for the two compounds and
agree with Köhler predicted activation values for AS and SA. Doubly charged aerosols are indicated in Figure 2 and
are a negligible contribtion to the activation curves.

The supersaturation and critical dry diameter are related via the single parameter hygroscopicity, **$\kappa$,** to describe

the CCN activity and to determine the effect of mixing states of multiple components on the supersaturated
hygroscopic properties of aerosols. Using the generalized $\kappa$-Köhler equations presented by Petters and Kreidenweis
(2007 and 2008), droplet growth in the supersaturated regimes for a selected dry diameter can be modeled for an
aerosol where the entire particle diameter dissolves at activation.

$$ln S_c = \left( \frac{4A^3 \rho_w M_s}{27 v \rho_s M_w D_d^3} \right)^{1/2}, \quad where \ A = \frac{4\sigma_{s/a} M_w}{RT\rho_w} \qquad (1)$$


$$\kappa = \frac{4A^3}{27 D_d^3 ln^2 S_c} \quad (2)$$


$\sigma_{s/a}$ is the surface tension, $M_w$ is the molecular weight of water, $R$ the universal gas constant, $T$ is the temperature at
activation, and $\rho_w$ is the density of water. Surface tension and density of water were calculated according to
temperature dependent parameterizations presented by Seinfeld and Pandis (1998) and Pruppacher and Klett (1997).
The surface tension of the solution is assumed to be that of pure water.  Traditional Köhler theory is known to work
reasonably well for inorganic salts and slightly-soluble and hygroscopic organics like succinic acid.

## 4. Results and Discussion

### 4.1 Internal Mixtures

Aerosolized internally mixed solutions exhibit single CCN activation curves for all five compositions of succinic acid and NaCl solutions (Fig. 3a). The activation curve is similar to that of ammonium sulfate in Figure 2a. Multiply charged particles contribute less than 10% to the activated fraction and are assumed to be negligible. A sigmoid that plateaus near one can be applied to describe the CCN activation. As the internal mixture salt fraction increased at a given supersaturation, the single curve was maintained and shifted towards a lower activation diameter, indicative of and consistent with more hygroscopic aerosol. The shifting of the CCN activation sigmoid (left and right) is also expected of internally mixed particles formed via the coagulation of separate particle distributions (Farmer et al., 2015). Using the simple mixing rule, a multicomponent hygroscopicity parameter can be theoretically applied based on the expected kappa values for each individual component hygroscopicity ($\kappa_i$), and the volume fraction of each component ($\varepsilon_i$) (Petters and Kreidenweis (2007).

$$\kappa = \sum_i \varepsilon_i \kappa_i \qquad \textbf{(3)}$$

Equation (3) was applied for each mixture. $\kappa$ was calculated and compared to the experimental values (Fig. 3b). These internal mixtures do not strongly follow the simple mixing rule for every mixture and is consistent with the previous work of Shulman et al. (1996) and Padró et al. (2007) who showed that slightly soluble compounds internally mixed with salts resulted in surface tension depression and thus a lower required critical supersaturation. As previously published, accounting for organic surface tension depression could improve kappa-hygroscopicity calculations for internal mixtures. Regardless of surface tension omissions, the single sigmoidal experimental data set is within 20% of theoretical agreement and indicative of a single component or homogeneous internally mixed aerosol population.

**Figure 3. a)** Activation curves for two component internal mixtures of NaCl and succinic acid (SA) at SS 0.72%. Doubly charged aerosols are present but are all below 0.1 and negligible. **b)** Internally mixed aerosols.

Multicomponent hygroscopicity parameter predictions vs. experimentally derived kappa values. Dashed lines

indicate 50% uncertainty (1:1.5 and 1.5:1). Data are within 20% uncertainty of the 1:1 line.

**4.2 External to Internal Aerosol Mixing Results**

Individual single component aqueous solutions of ammonium sulfate (AS), $(NH_4)_2SO_4$, and succinic acid

(SA) were aerosolized, dried with an active heated column and silica gel diffusion dryer to produce external
mixtures. Data sets yielding multiple activation curves consistent with external mixing were successfully created by
*i)* mixing aerosol streams and *ii)* injecting SA and AS-compounds in the flow tube. For this manuscript, we show
externally mixed data generated from the use of the mixing flow tube. The direct mixing produces the same external
mixed CCN results (not shown). By using aerosols consisting of compounds of significantly different
hygroscopicities, and thus different activation diameters, distinct double plateaus for CCN activation can be
observed for external mixtures (Fig. 2b). At particle mobility diameters between ~35 and 45 nm there is an
asymptote, $\eta \sim 0.6$ (Fig. 4a., 4b., and 4c). It is noted that heated succinic acid particles can evaporate during aerosol
generation before CCN measurement; the asymptote in Figure 4 is an indication that multiple components are
present in the total aerosol distribution.  The activation curves were characteristic of AS and SA, and the measured
activation diameters agreed well with Köhler Theory and the single parameter ($\kappa$) thermodynamic predictions of
droplet activation (Fig. 4a). The external mixture was maintained for an hour as indicated by the separate and stable
activation diameters derived from multiple sigmoid analysis. For more than 2-component externally mixed particle
distributions,  more than two plateaus will be observed. For example, the work of Schill et al. 2015 shows  multiple
plateaus for a five-component external mixture mimic of ocean spray CCN.  If one considers a limiting case of
infinite components with distinct varying degrees of hygroscopicity, then the activation curve will be monotonic
below 1 and may appear to be representative of an internal mixture; the CCN activation curve will approach the
shape of a shallow sigmoidal slope (however not as steep or instantaneous as the ideal step function).

**Figure 4. External to internal mixtures of a slightly soluble organic aerosol, succinic acid (SA) and inorganic**
**ammonium sulfate (AS) aerosol.** The CCNC was operated at a single $S_c$ of 0.8%. Closed symbols are externally

mixed.  Open symbols are internally mixed. a) The apparent kappa values derived from externally mixed multiple-

sigmoid activation data before active heating is turned off (at approximately 3:30 pm) and single sigmoid activation

curves (after 3:30pm) are shown. The two dotted lines indicate the theoretically derived kappa values for succinic

acid, $\kappa$=0.23, and ammonium sulfate, $\kappa$=0.61. b) CCN activation curves exhibit external (with active heating before

3:30 pm) and then transitioning external to internal mixing (after 3:30pm) c) CCN/CN vs. Dry mobility diameter

data as a function of time.  The asymptote, $\eta$, disappears at the by the end of the experiment.


One hour after initial injection into the flow tube, the active heating column was turned off. It should be

noted that atomized aerosol continued to be dried through the silica gel diffusion dryers, as is commonly done. The
relative humidity after the dryer in both cases is small ($< 20\%$) and thus the activation diameters of very hygroscopic
AS calibration aerosol are not affected with or without active heating (Fig. A2).  However, as soon as active heating
was turned off, particles in the mixing flow tube became more mixed (Figure 4).  Thus, it is likely that minute
amounts of aerosol water promoted internal mixing and shifted aerosol mixing from external to internal in the
mixing flow tube system.

CCN activation curves for the two compounds remained distinct and separate until internal mixing

conditions dominated and the multiple CCN activation curves converged into a single curve (Fig. 4b and 4c).
Results suggest aerosol water plays a significant factor in mixing and CCN activation. This is consistent with
previous work that indicates that the presence of water led to lowered aerosol viscosity and increased diffusivity (Ye
et al., 2016). The wetted aerosols can come in contact through diffusion and coalesce to form an internally mixed
aerosol. The apparent kappa values from fitting the two individual activation curves for the external part of the
mixing experiment and subsequent internal mixing are shown in Fig. 4a.

To help track the change in organic/inorganic fractions during the transition from external to internal, the

mixed aerosols were analyzed with a high-resolution time of flight mass spectrometer (HR-Tof-AMS) to provide
mass fraction information. The mass size distribution was integrated and normalized for each compound per scan
according to the total mass that was measured. The mass size distribution was then converted to number size
distribution and the diameters were converted from aerodynamic diameter to electrical mobility diameter. Then for
each superstation and fraction, the EMF was calculated between the two respective activation diameters and
correlated to the EMF that was determined from SMCA to determine the plateau height ($\eta$).

AS fractions measured by the AMS are consistent with the changes SMCA derived $\eta$ where increases in

AS mass fraction increase $\eta$.  However, the AMS derived AS fraction are slightly lower compared to SMCA (Fig.
5), indicating potential influence on $\eta$ from other factors. Previous work has shown the presence of highly soluble
materials (like AS) can promote CCN activity of organic dominated systems (Asa-Awuku et al., 2011; Fofie et al.,
2017). A second flow tube experiment was conducted to test the effect of differing concentrations on plateau
heights. The detection efficiency for particles with smaller sizes in the AMS (<50nm) can effect the AS fraction.
Thus, the CCNC supersaturation was modified from 0.2 to 1.2% to test the effect of activation diameter on closure
(Fig. 6.). Results show good agreement and are within 50% of predictions. Results with distinct CCN activation
plateaus, especially of 2-3 distinct hygroscopicities, may be useful for estimating aerosol chemical fractions when
other instruments with lower size resolution are not readily available.


**Figure 5.** Plateau heights derived from AMS data vs. SMCA. Single CCNC Supersaturation. Dashed lines
indicate 50% uncertainty (1:1.5 and 1.5:1).


**Figure 6.** Modifying activation diameters. Plateau heights derived from AMS data vs. SMCA for CCNC
Supersaturation from 0.2 to 1.2%. Dashed lines indicate 50% uncertainty (1:1.5 and 1.5:1).


**4.3 External and Internal Mixtures of Combustion Aerosol**
Combustion aerosol or soot can form external and internal complex aerosol mixtures. Soot is considered
insoluble but wettable (e.g., but not limited to Lammel and Novakov 1995; Moore et al. 2014) and the contributions
of Black Carbon containing particles to aerosol-cloud interactions at varied mixing states is not well known or
understood (Lammel and Novakov 1995; Novakov and Corrigan 1996; Weingartner et al. 1997; Dusek et al. 2006;
Kuwata and Kondo 2008; Koehler et al. 2009; Bond et al. 2013; McMeeking et al. 2011; Liu et al. 2013; Rojas et al.
2015). Thus, the ability of black carbon to mix with inorganic and organic compounds and to observe the extent of
mixing as they activate as CCN is of great interest.
Prior to investigating the impact of mixing fresh combustion emissions with inorganic and organic aerosols, the
CCN activation spectra of soot was measured using combustion aerosol generated from the APG. The aerosol is

319 likely composed of black carbon and oxidized carbonaceous material (Moore et al. 2014; Durdina et al. 2016). Thus

320 we also refer to carbonaceous aerosol as black carbon mixtures (simply, BC mixtures). The CCN activated fraction

321 data from soot was fit to a singular sigmoidal curve (Fig. 7). There are no plateaus in the activation curve and the

322 single sigmoid fit indicates that the aerosol generated is a homogenous internal mixture.  Combustion aerosol

323 activated at a mobility diameter of 133 nm at 2.2% supersaturation. The apparent hygroscopicity of combustion

324 aerosol was $\kappa$=0.001, and is consistent with the order of magnitude and kappa values reported for fresh combustion

325 aerosol from diesel engine sources (Fig. 7) (Vu et al., 2015; Moore et al. 2014). It is noted that the apparent

326 hygroscopicity is defined by the electrical mobility diameter that assumes particles are spherical.

329    **Figure 7.** Combustion aerosol activation curve (SS 2.2%, $d_{p,50}$=133nm, $\kappa$=0.001)

330  Next, the influence of modifying externally mixed hygroscopic aerosol fractions with non-hygroscopic BC

331 mixtures was observed. Soot was externally mixed with various concentrations of AS and NaCl in two separate

332 experiments. For each BC mixture, combustion aerosol was introduced to the flow tube and atomized inorganic and

333 organic aerosol (dried with a heated column and silica gel diffusion dryers) was injected.  The concentration of

334 inorganic in the flow tube was slowly increased to modify the contribution of soluble material. The CCN counter

335 supersaturation was decreased to 1.1% to observe the impact of the more hygroscopic compounds in the BC

336 mixture.

337  Figure 8 shows the CCN activation of external mixtures of BC with AS and NaCl at $S_c$ = 1.1%. The initial

338 combustion size distribution at the start of the experiment and the modified CCN activation fractions of the aerosol

339 mixtures are presented. The shape of the activation curve provides insight about the sizes of very hygroscopic and

340 non-hygroscopic species. At $S_c$ = 1.1% BC particles, with $\kappa$ = 0.001, will not theoretically activate below 250 nm

341 and the contribution of externally mixed BC in the size range shown does not contribute much (if any) CCN. The

342 normalized size distribution data show that there are few BC-like particles at small sizes (<50 nm), the majority of

343 particles in this range are inorganic. Thus for both the AS and NaCl external mixtures, the activation diameters

344 derived from a singular fit were consistent with the expected $d_{p50}$ < 50 nm of the respective inorganic salts.

345 Specifically at $S_c$=1.1%, AS and NaCl particles activated at  25 and 19 nm  respectively (congruent with theoretical

346 dp50 at 24.8 and 19.0 nm) At the larger sizes (> inorganic $d_{p50}$), the BC mixture concentration increased and the

CCN/CN was depressed. The combustion aerosol alone is not CCN active at this $S_c$ or size and the depressions are
reflective of the non-hygroscopic combustion aerosol fraction in the aerosol sample. Notably, plateaus are dynamic.
As the concentration of inorganic salts increase, the increased activated fraction is reflected in the CCN spectra; the
plateau heights increase with increasing hygroscopic concentrations. In these particular externally mixed
experiments, the initial CCN/CN plateau can be as large as one, subsequently decrease, and then will likely increase
to one after the BC critical diameters are reached. BC externally mixed with very hygroscopic material is more
CCN active than the soot alone.


**Figure 8.** Combustion aerosols externally mixed with inorganics a) NaCl externally mixed with concentrations
modified from 51% to 85% over the course of 60 min b) AS externally mixed with concentrations from 41% to 86%
over a course of 75 min. Cross symbols represent the initial size distribution of the combustion aerosol.

Succinic acid (SA) was mixed with combustion aerosol to investigate the external to internal mixing and transition
of slightly hygroscopic organic with non-hygroscopic insoluble but wettable aerosols. The laboratory system mimics
observed increases in SOA mass fractions on combustion aerosols during atmospheric aging. The SA was
introduced to the flow tube at various concentrations, followed by the combustion aerosol from the APG, under dry
externally mixed conditions and bimodal size distribution peaks were observed (Figure 9).

The normalized size distributions of the aerosol leaving the flow tube are presented. The initial soot size distribution
is similar to those presented in Figure 8. Assuming the first of the two peaks is SA, and the second is a mixture of
the combustion aerosol and SA, the initial point of activation agrees with that of succinic acid where SA aerosols all
activate. After a mobility diameter, $dp$ > 50 nm, the concentration of combustion aerosol in the mixture increases
and the CCN/CN ratio is < 0.2, indicative of a lower SA concentration relative to the non-activated BC
concentration.

To induce internal mixing, active heating was again turned off for the atomized aerosol source. Again, internal
mixing was promoted and the multiple activation curves converge into a single sigmoid for the BC and SA system.
This is consistent with the AS/SA experiment and previous work that showed a strong influence of insoluble
compounds on activation when internally mixed with a more soluble compound. (9) With continued mixing, a shift
to larger activation diameters was observed towards the end of the experiment (scan 93) and there was a slight
depression in the plateau of the CCN spectra.

The data suggests that only a small amount of soluble inorganic and organic material is required to make the soot
more active than that observed alone, especially as the aerosol becomes more internally mixed.


**Figure 9.** a) Time series of CCN/CN activated fractions of Succinic Acid (SA) and combustion aerosol (BC)
mixture in flow tube. b) The CCN/CN activated fraction (closed triangles) of SA and BC mixtures for particle
distribution scans 24, 45, 54, 73, and 94. Aerosol water is introduced at ~ scan 70 to promote internal mixing. Cross
symbols show the particle size distribution mixed aerosol.

## 5. Conclusion
Results confirm that the experimental CCN activation curves of aerosol provide insight into to the type of
mixing (e.g. internal vs. external) and the various levels of hygroscopicities that are chemically representative.
Modifications in the concentrations of externally mixed non-hygroscopic aerosols are reflective of the CCN
activated concentrations. This is consistent with CCN spectra observed in ambient studies, which have attributed
observations in CCN activation plateau heights less than one to the contributions of externally mixed and inactivated
(typically non-hygrosocopic black carbon) aerosols. This work adds to the existing body of CCN literature and
demonstrates that the transition from external to internal mixtures can be mimicked in controlled laboratory
experiments and observed in CCN data. If one accounts for multiple-sigmoid analysis in experimental CCN
activation data, the CCN behavior of known hygroscopic compound mixtures (e.g., ammonium sulfate, sodium
chloride, succinic acid) agrees well with traditional Köhler theory. However, more work is needed to explore
mixtures of hygroscopic material particularly with wettable and non-hygroscopic aerosol. Here, as the non-
hygroscopic combustion aerosol becomes internally mixed with the inorganic and organic material, the CCN activity
of the combustion aerosol is modified. The data here, with recent publications (Altaf et al., 2018; Ye at al., 2016),
suggest that aerosol water is a significant factor in promoting mixing and can be used to modify mixing states. To
our knowledge, the technique provided here is the first to show aerosol population transitions from external to
internally mixed states in a laboratory environment and thus the technique can be applied to understand additional
aerosol properties (e.g., optical, gas-phase uptake, subsaturated droplet growth, etc.) of known compounds that can
modify particle mixing states.  The aerosols studied here maintained an external mixture under dry conditions; CCN
activation curves plateaued and remained constant. However, turning off the heater promoted internal mixing and
the activation curves were observed to converge. These experiments with known compounds validate that the
aerosol mixing state can be observed in CCN activation data and can be applied to aerosol data sets to understand
the extent of mixing. The results confirm that CCN counters and CCN analysis should be used in future studies to
quantify the extent of mixing of ambient particles.  The results are critical to understanding other factors important
to the direct and indirect radiative contributions of atmospheric particles.

## Acknowledgments

This work was supported by the University of California Transportation Center and the U.S. Environmental
Protection Agency (EPA) grant number 83504001. Diep Vu thanks the U.S. Environmental Protection Agency
(EPA) STAR Fellowship Assistance Agreement no. FP-91751101. EPA 83504001 was fundamental for the
development of the mixing flow tube apparatus. Additionally, AA would like to thank the National Science
Foundation (NSF) proposal 1151893. The contents of this paper are solely the responsibility of the grantee and do
not necessarily represent the official views of the EPA or NSF.  Further, the EPA and NSF do not endorse the
purchase of any commercial products or services mentioned in the publication.   In addition the authors would like
to acknowledge Desiree Smith, Drs. Kent Johnson and  Heejung Jung for their role in the acquisition of APG  and
access to controlled  BC measurement and Dr Jeffrey Pierce for advice on CCN models.

**Data availability.**

The data and figures will be made available upon request from authors Diep Vu and corresponding author Akua Asa-Awuku.

**Author contributions.**

SG built the mixing flow tube device used in the experiments. KV developed fluid models to characterize the flow of particles in the instrument. TB and MB assisted with experimental measurements presented. DV conducted measurements, analysis and contributed to the writing of the manuscript. AAA developed the concept for this work and contributed to the analysis and writing of the manuscript.

**Competing interests.**

The authors declare that they have no conflict of interest.

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

Number of Pages: 14

Number of Figures: 9

Number of Appendix Figures: 2

* Corresponding Author.

*E-mail address:* asaawku@umd.edu (A. Asa-Awuku)

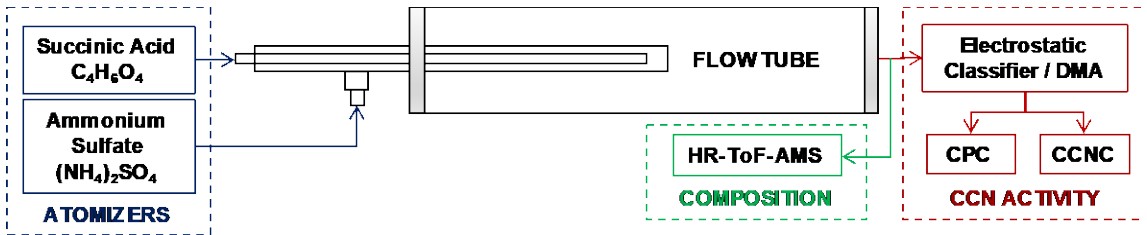

fig01

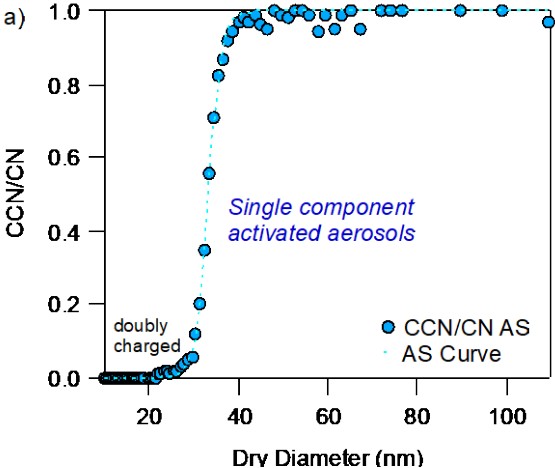

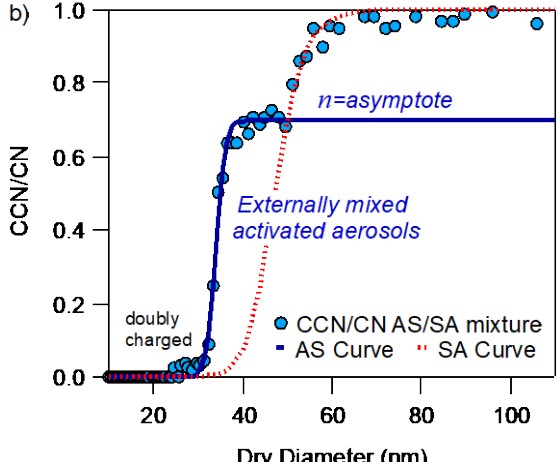

fig02

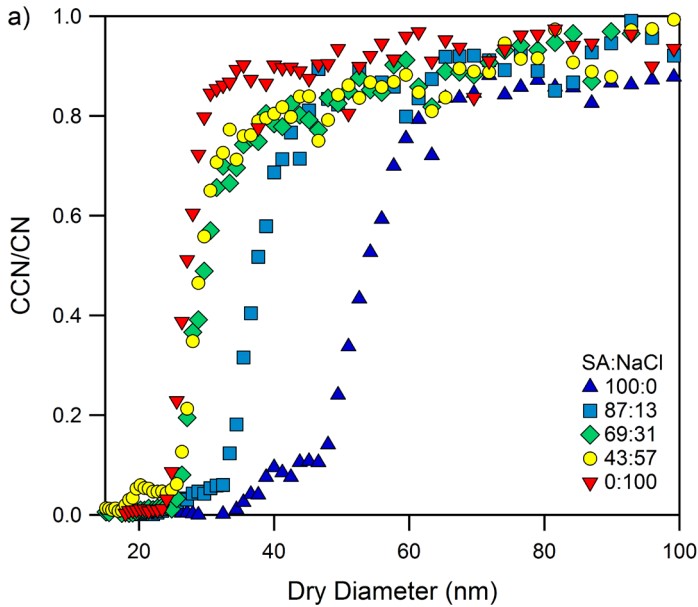

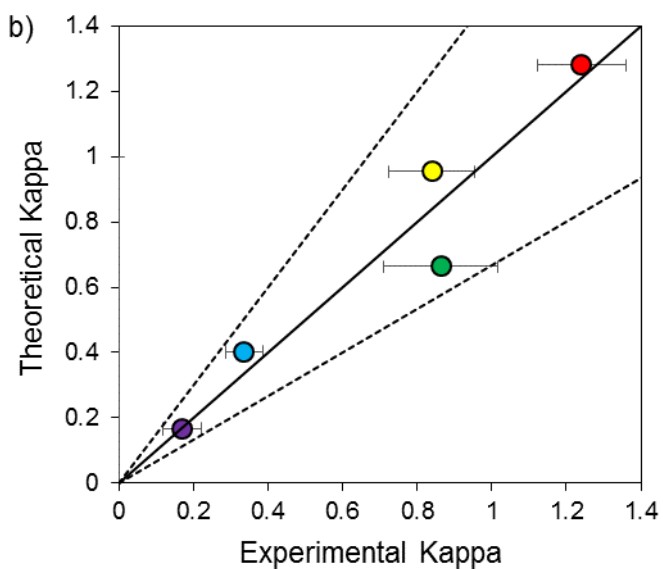

fig03

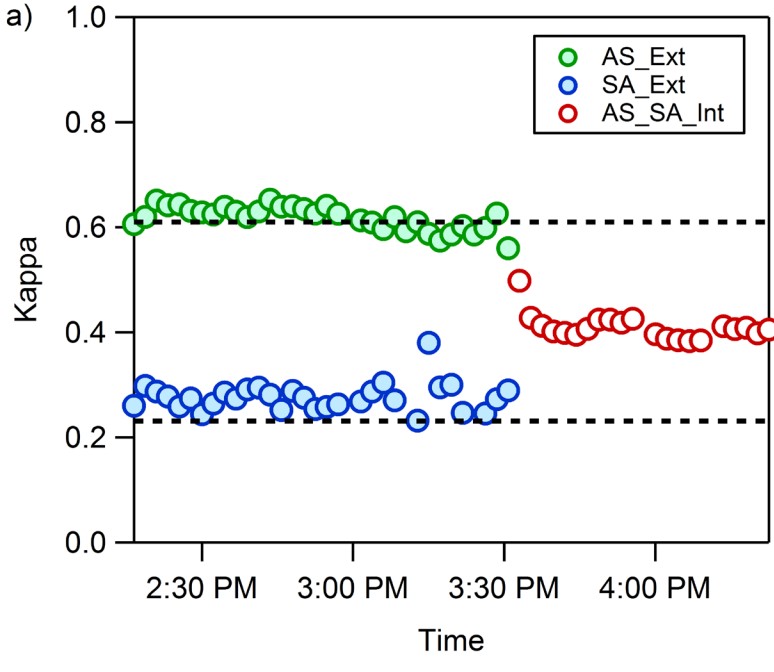

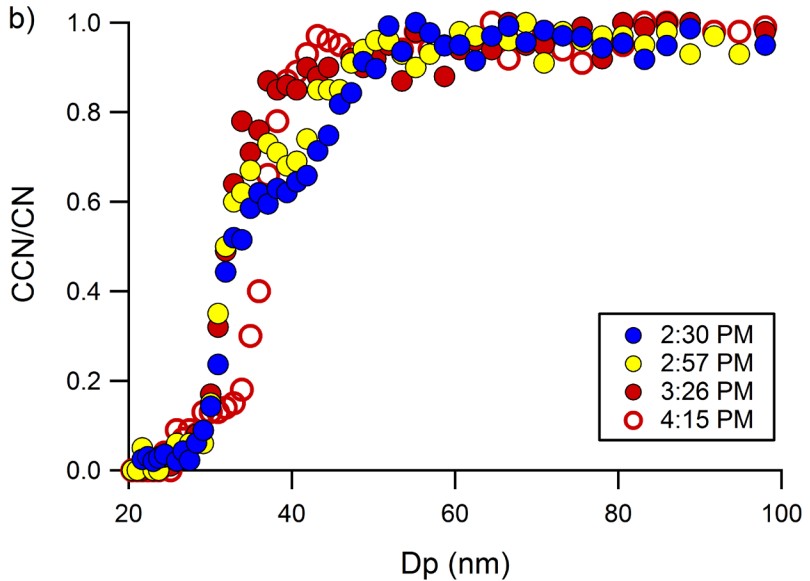

c)

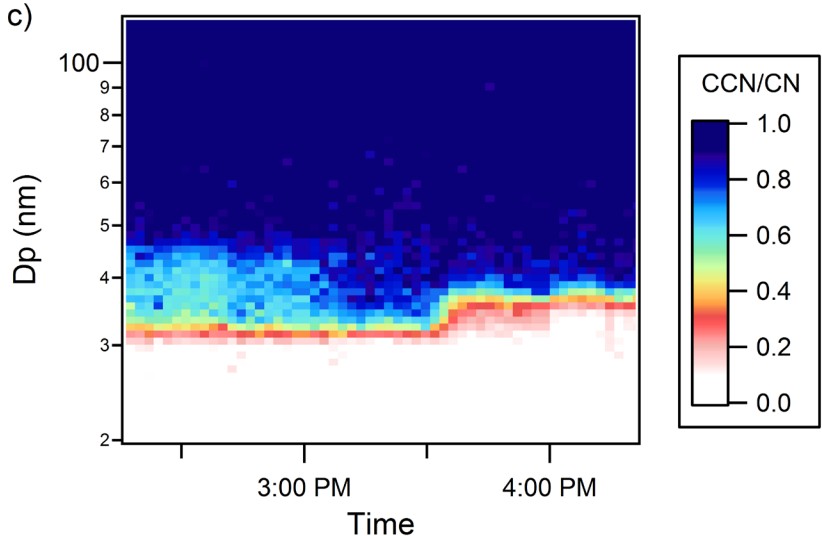

fig04

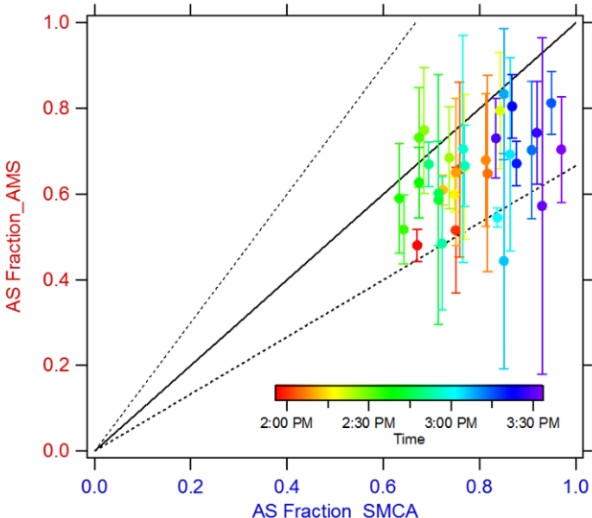

Fig05

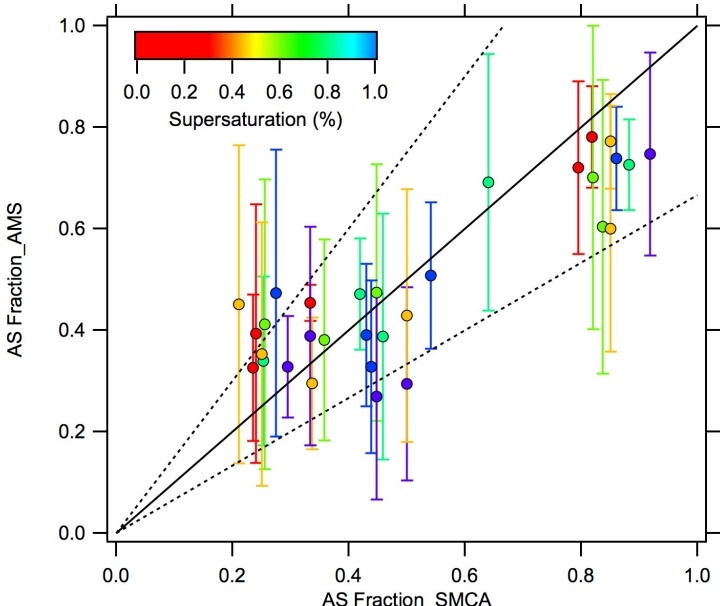

fig06

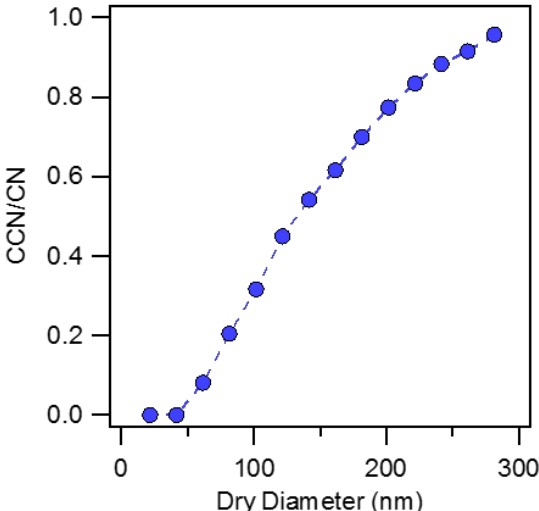

fig07

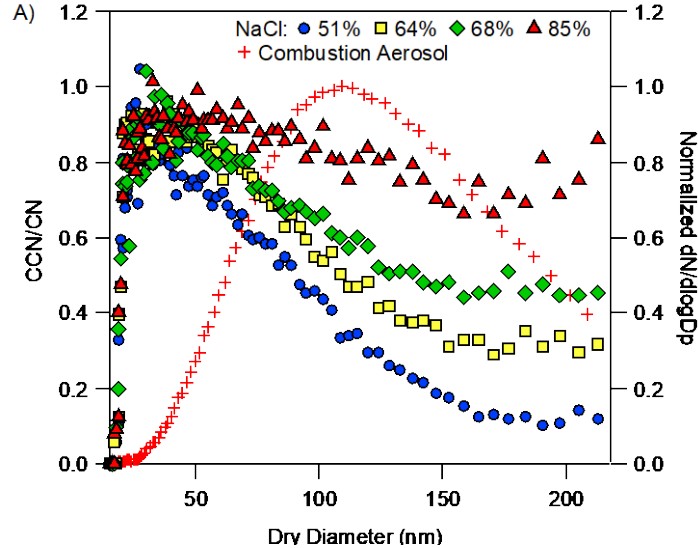

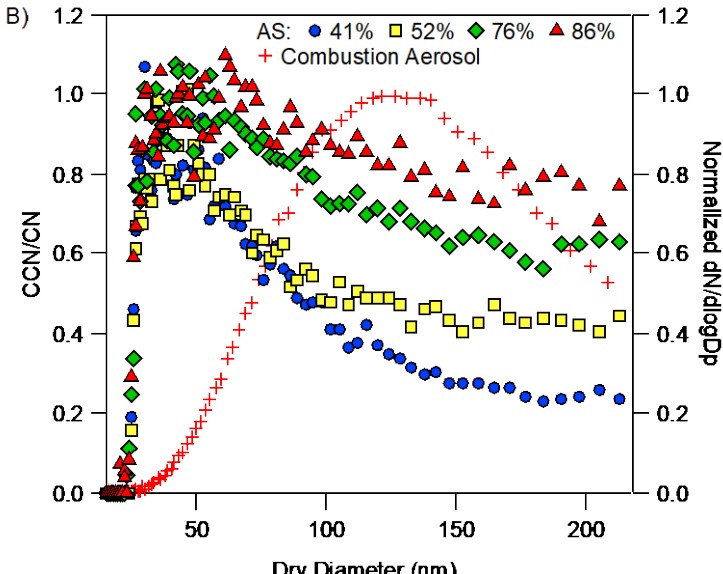

fig08

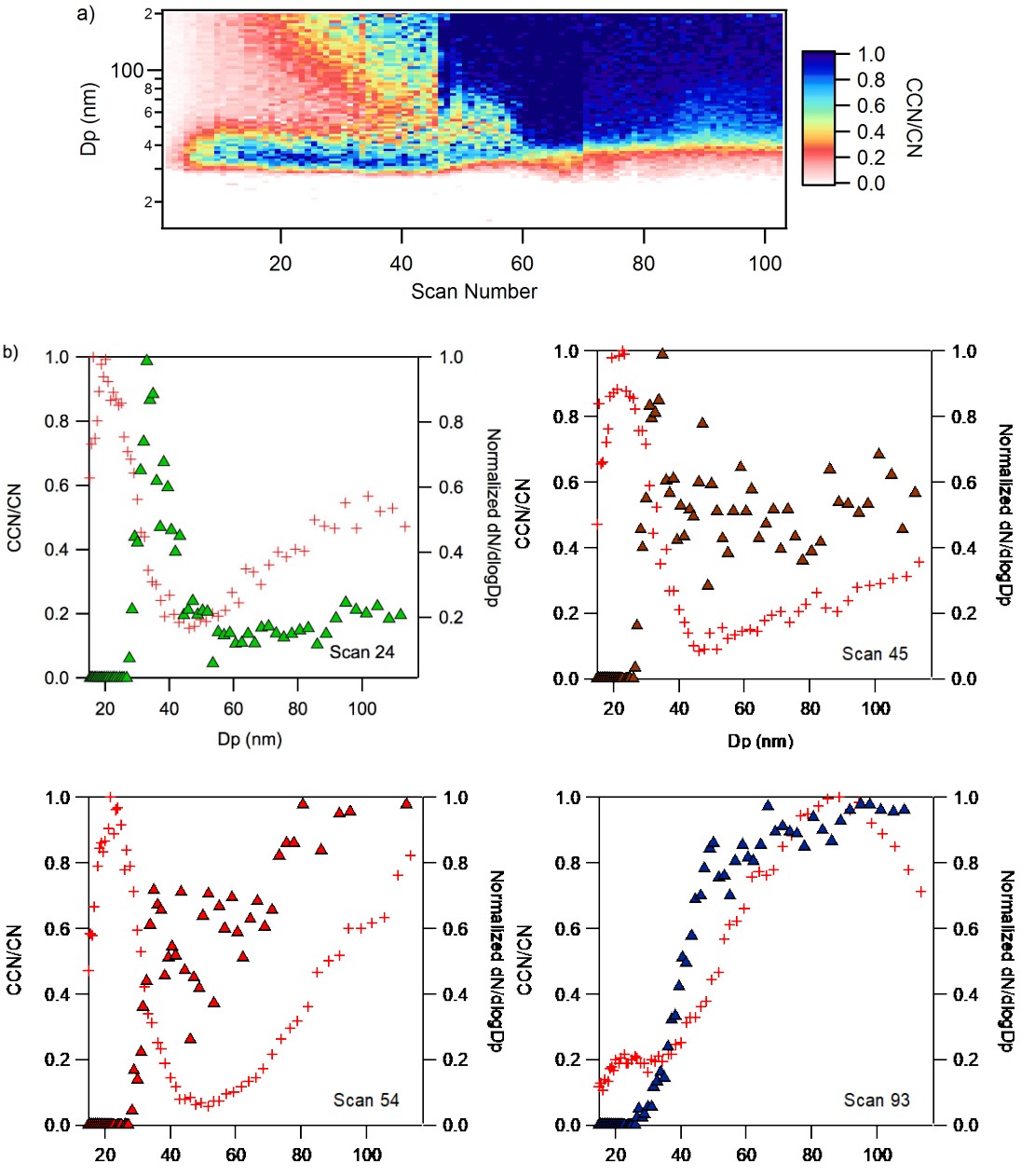

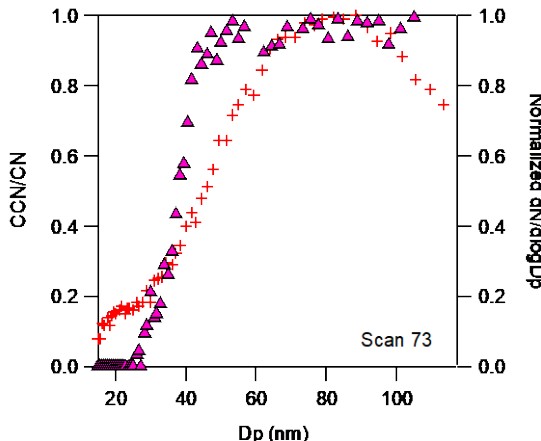

Fig09