# Peer review of "External and Internal Cloud Condensation Nuclei (CCN) Mixtures: Controlled Laboratory"

_Atmospheric Measurement Techniques, 2018_

## Referee Comment (RC1) · Anonymous Referee #2 · 16 Feb 2019

The manuscript by Vu et al. discusses about the experimental method and data analysis for CCN activity of externally mixed particles. Although the technical approach sounds reasonable, the reviewer is skeptical about the novelty. I suggest the authors to significantly improve the manuscript by appropriately referring other papers so that the readers will be able to understand the significance of the manuscript better.

Major comments:

L68 'However, dynamic changes in particle mixing states and subsequent treatment of CCN measurement and analysis have not been readily observed and studied in depth.'

I disagree with the statement. A number of papers has been published about the topic in the last decade ( e.g., Kuwata and Kondo 2008; Su et al. 2010). I suggest the

authors to conduct intensive literature search carefully again.

L78 'BC is renown for its direct radiative effects yet little is known experimentally about the contributions of BC to aerosol-cloud interactions at varied mixing states.'; L283' the contributions of BC to aerosol-cloud interactions at varied mixing states is not well known or understood'

Even though there might be only a limited number of studies, some researchers have conducted laboratory experiments/atmospheric observations on CCN activity of BC particles. I suggest the authors to conduct literature search carefully again. For example, there are some description about it in a review paper of BC particles (Bond et al. 2013).

Minor comments:

Title 'CCN Mixtures'

It is not clear to me how this word is defined. Please clarify it, or consider to use other expressions.

L60 'the CCN mobility diameter data sets'

What does this word mean? Please clarify it.

L107 'Five aqueous solutions of succinic acid and NaCl with molar ratios of 100:0, 87:13, 69:31, 43:57, 0:100 were aerosolized using a single atomizer'

I wondered if potential evaporation of succinic acid in the experimental setup could influence the final molar ratios of the compounds. Do the authors have any comments on it?

L112 'As conditions (e.g., but not limited to, residence time, temperature, pressure, relative humidity) change in a flow tube,'

I wondered how well these parameters were controlled during this study. Please clarify.

[Figure]

L170 'For this study the first curve is similarly defined as the hygroscopic externally mixed fraction (EMF) with an asymptote, or plateau of $\eta$. The dependence of $\eta$ varies with the presence of mixed components and their respective hygroscopicities. Thus we evaluate $\eta$ for controlled compositions and compare how representative they are of the known fractions of mixtures. '

It may be a good idea to add a figure to explain about the concept.

L241 'After one hour in the flow tube'

Does it mean that particles stayed in the flow tube for one hour? Please clarify.

L257 'The mass size distribution was then converted to number size distribution'

How does the error/uncertainty for measuring mass-size distribution influence the estimated number size-distribution? It would be ideal to conduct sensitivity study about it.

L282 ' Combustion Aerosol, hereon also referred to as Black Carbon (BC)'

Combustion aerosol is not equivalent as black carbon.

L283 'BC is considered insoluble but wettable'

Please show experimental data/cite literature to support the idea.

L288 'The single sigmoid fit suggests that the aerosol generated is a homogenous mixture of black and brown carbon.'

The activation curve is very broad. I do not think that such a broad activation curve could be observed if all the particles have exactly the same chemical composition. I wonder what the authors wanted to mean by 'homogeneous.' How can the authors make sure that brown carbon existed in the particles they measured? Please demonstrate the evidence for it.

L292 'It is noted that the apparent hygroscopicity does not account for non-spherical

fractal particles.'

I do not understand what this sentence means. Do the authors want to clarify that these particles do not have spherical shape?

References: Bond, T. C., Doherty, S. J., Fahey, D. W., Forster, P. M., Berntsen, T., DeAngelo, B. J., Flanner, M. G., Ghan, S., Karcher, B., Koch, D., Kinne, S., Kondo, Y., Quinn, P. K., Sarofim, M. C., Schultz, M. G., Schulz, M., Venkataraman, C., Zhang, H., Zhang, S., Bellouin, N., Guttikunda, S. K., Hopke, P. K., Jacobson, M. Z., Kaiser, J. W., Klimont, Z., Lohmann, U., Schwarz, J. P., Shindell, D., Storelvmo, T., Warren, S. G., and Zender, C. S. (2013). Bounding the role of black carbon in the climate system: A scientific assessment. J. Geophys. Res. 118:5380-5552.

Kuwata, M. and Kondo, Y. (2008). Dependence of size-resolved CCN spectra on the mixing state of nonvolatile cores observed in Tokyo. J. Geophys. Res. 113:D19202.

Su, H., Rose, D., Cheng, Y. F., Gunthe, S. S., Massling, A., Stock, M., Wiedensohler, A., Andreae, M. O., and Poschl, U. (2010). Hygroscopicity distribution concept for measurement data analysis and modeling of aerosol particle mixing state with regard to hygroscopic growth and CCN activation. Atmos. Chem. Phys. 10:7489-7503.

---

## Referee Comment (RC2) · Anonymous Referee #3 · 15 Apr 2019

The manuscript represents a method of CCN data analysis for different mixtures of organic and inorganic components of varying mixing states with regards to the observed activation and hygroscopicity. By conducting controlled laboratory experiments of mixing, the transition from external to internal mixtures are effectively mimicked. For mixtures of known hygroscopicity CCN behavior agrees well with traditional Köhler theory. Finally, aerosol water is shown to play a significant role in promoting mixing and can be used to modify mixing states.

The paper is well written and easy to follow, though there are some issues and more thorough discussion should be made in specific sections. A very interesting point of the study is the deconvolution of the activation curves to different sigmoidal curves consistent with those of the different components, as well as the study of the impact of

mixing fresh combustion emissions with inorganic and organic aerosols.

Specific comments:

1) More thorough discussion should be made in the study of mixing fresh combustion aerosol with inorganic and organic aerosols. Why is combustion aerosol considered and referred to solely as Black Carbon? Also, the activation curve presented in Fig.07 does not reach an obvious plateau, how can we speak about a "homogeneous mixture of black and brown carbon"? Also I would expect that as a homogenous mixture, the activation curve would be a lot steeper, especially in the larger particle size range. Can you please clarify/comment on this?

2) A more thorough review of the current literature should be made, as there are relevant studies that are not mentioned. E.g. L69-70 Moore et al., 2012 provide an in-depth analysis of ambient CCN measurements during the 2010 Deepwater Horizon Oil Spill, both studying hygroscopicity vs mixing state and type of size distributions and droplet activation kinetics.

3) CCN activity also depends on the atmospheric processing as well. L50-51 it is stated that CCN activity is complicated when inorganic salts are internally mixed with a complex organic. It has been shown in filed studies that even within a few hours of processing, hygroscopicity (thus activity as well) becomes more or less constant, e.g. Lathem et al., 2013; Bougiatioti et al., 2009. This, to my opinion, should be mentioned.

Technical corrections:

L109 What type of SMPS, sizing range and of CCNc? It is mentioned in Section 2.3. but it seems to me as missing information to me at that early stage.

L293 What do the authors mean by "does not account for non-spherical fractal particles"?

Fig.05 Some points exhibit huge variation

[Figure]

---

## Author Comment (AC1) · 10 May 2019

The manuscript by Vu et al. discusses about the experimental method and data analysis for CCN activity of externally mixed particles. Although the technical approach sounds reasonable, the reviewer is skeptical about the novelty. I suggest the authors to significantly improve the manuscript by appropriately referring other papers so that the readers will be able to understand the significance of the manuscript better.

Major comments:

L68 'However, dynamic changes in particle mixing states and subsequent treatment of CCN measurement and analysis have not been readily observed and studied in depth.' I disagree with the statement. A number of papers has been published about the topic in the last decade ( e.g., Kuwata and Kondo 2008; Su et al. 2010). I suggest the authors to conduct intensive literature search carefully again.

→ We have added the suggested references.

→ We have modified the sentence to read "However, dynamic changes in particle mixing states and subsequent treatment of CCN measurement and analysis have not been readily observed and studied in depth **under controlled laboratory conditions".**

→ We agree with the reviewer that there have been many studies (particularly field studies) that have explored changes in aerosol mixing state and CCN. Kuwata and Kondo 2008 measured ambient air in Tokyo and infer the mixing state properties of the ambient aerosol with subsequent behaviour on CCN. Su et al. 2010 brilliantly develop theoretical models to describe the CCN activation of mixed population and then validate the models with aerosol measurements taken in Beijing, China. A thorough literature review will show that the majority of these papers (like the ones suggested) apply mixing state assumptions to ambient data sets. To our knowledge, we are the only manuscript to show changes in laboratory CCN with controlled *dynamic* changes in mixing state aerosol population (*i)*from external to internal mixtures and *ii)* not to be confused with aerosol-phase states or changes in aerosol coatings). This is the novelty of the presented technique. Previous papers (dating back to the early 70's) have provided appropriate mixing, size-resolved composition theory and have shown the activation of laboratory externally mixed aerosol and internally mixed systems, separately. We have yet to find a CCN paper that has controlled and measured the dynamic change from external to internal mixtures in the laboratory. We apply the technique to CCN but the method can also be extended to understand optical, sub-saturated, and other physicochemical characteristics. We regret that this uniqueness did not come through in the first revision of the paper and thus we have spent considerable effort emphasizing the importance of the new technique presented in the revised manuscript.

→ The following text has been revised to include additional references as follows.

Knowledge of the mixing state and the chemical composition can greatly improve CCN predictions and has been the focus of several studies (e.g., but not limited to (Bilde and Svenningsson 2004; Abbatt et al. 2005; Henning et al. 2005; Svenningsson et al. 2006; King et al. 2007; Cubison et al. 2008; Kuwata and Kondo 2008; Zaveri et al. 2010; Su et al. 2010; Wang et al. 2010; Spracklen et al. 2011; Ervens et al. 2010; Asa-Awuku et al. 2011; Liu et al. 2013; Jurányi et al. 2013; Paramonov et al. 2013; Padró et al. 2012; Moore et al. 2012; Meng et al. 2014; Bhattu and Tripathi 2015; Almeida et al. 2014; Schill et al. 2015; Crosbie et al. 2015; Che et al. 2016; Ching et al. 2016; Mallet et al. 2017; Sánchez Gácita et al. 2017; Cai et al. 2018; Schmale et al. 2018; Mahish et al. 2018; Kim et al. 2018; Chen et al. 2019; Stevens and Dastoor 2019)

L78 'BC is renown for its direct radiative effects yet little is known experimentally about the contributions of BC to aerosol-cloud interactions at varied mixing states.'; L283' the contributions of BC to aerosol-cloud interactions at varied mixing states is not well known or understood' Even though there might be only a limited number of studies, some researchers have conducted laboratory experiments/atmospheric observations on CCN activity of BC particles. I suggest the authors to conduct literature search carefully again. For example, there are some description about it in a review paper of BC particles (Bond et al. 2013).

→ We have updated the references and provided the following text.
→Previous work investigating the contribution of BC to aerosol-cloud interactions at various mixing states has been studied (e.g., but not limited to Bond et al., 2013 and references therein, (Lammel and Novakov 1995; Novakov and Corrigan 1996; Weingartner et al. 1997; Dusek et al. 2006; Kuwata and Kondo 2008; Koehler et al. 2009; Bond et al. 2013; McMeeking et al. 2011; Liu et al. 2013; Rojas et al. 2015)

Minor comments:
Title 'CCN Mixtures' It is not clear to me how this word is defined. Please clarify it, or consider to use other expressions.
→ The title has been changed to Cloud Condensation Nuclei (CCN).
→ 'CCN mixtures' is simply a mixture of CCN.  It refers to some of the earliest work in the field by Winkler 1973 that accounts for multiple chemical compositions in aerosol. Specifically, this definition of aerosol mixture refers to a property of the overall particle population not to the property of an individual particle (Riemer et al., 2019; Winkler, 1973).  Recent studies have modified this definition to include changes in aerosol physical phase-state and also changes in individual particles. We have provided more detail in the revised manuscript  and have also added the recent and comprehensive review of Riemer et al 2019 to clarify in the text.

**References added**
→ Winkler, Peter. "The growth of atmospheric aerosol particles as a function of the relative humidity—II. An improved concept of mixed nuclei." Journal of Aerosol Science 4.5 (1973): 373-387.
→ Riemer, N., Ault, A. P., West, M., Craig, R. L., & Curtis, J. H. ( 2019). Aerosol Mixing State: Measurements, Modeling, and Impacts. Reviews of Geophysics, 57. **https://doi.org/10.1029/2018RG000615**

L60 'the CCN mobility diameter data sets' What does this word mean? Please clarify it.
→ The text has been revised to:
To help minimize the complexity in characterizing aerosol hygroscopic and CCN activation properties, CCN data analysis has traditionally been simplified by assuming that *i)* the aerosols share a similar or uniform hygroscopicity over a particle size distribution, *ii)* the CCN particle size can be described by the electrical mobility diameter, *iii)* CCN consists of few multiply charged aerosols and *iv)* all CCN active aerosols readily dissolve at activation.

L107 'Five aqueous solutions of succinic acid and NaCl with molar ratios of 100:0, 87:13, 69:31, 43:57, 0:100 were aerosolized using a single atomizer' I wondered if potential evaporation of

succinic acid in the experimental setup could influence the final molar ratios of the compounds. Do the authors have any comments on it?

→ We do not have a comment. Our data shows that the calculated molar solutions agree with predicted values to within 10% of their kohler prediction. Any potential evaporation that could occur is not reflected in the CCN values obtained from the internal mixture studies.

L112 'As conditions (e.g., but not limited to, residence time, temperature, pressure, relative humidity) change in a flow tube,' I wondered how well these parameters were controlled during this study. Please clarify.

→ The residence time, temperature, pressure, and relative humidity were conducted at ambient room conditions. Typical conditions were dry relative to any experiments conducted with aerosol water. Pressure and temperature were expected to be consistent from test to test (air conditioned laboratory with little temperature fluctuations, and minimal changes in barometric pressure). The above has been inserted and clarified in the text.

L170 'For this study the first curve is similarly defined as the hygroscopic externally mixed fraction (EMF) with an asymptote, or plateau of _. The dependence of _ varies with the presence of mixed components and their respective hygroscopicities. Thus we evaluate _ for controlled compositions and compare how representative they are of the known fractions of mixtures. ' It may be a good idea to add a figure to explain about the concept.

→ Please refer to figure 2. In the previous revision, we include this concept figure to explain the difference in plateaus of externally and internally mixed data.

L241 'After one hour in the flow tube' Does it mean that particles stayed in the flow tube for one hour? Please clarify.

→ this is in reference to 1 hr after initial injection. Text has been modified from 'After one hour in the flow tube' to 'One hour after initial injection into the flow tube' to clarify this.

L257 'The mass size distribution was then converted to number size distribution' How does the error/uncertainty for measuring mass-size distribution influence the estimated number size-distribution? It would be ideal to conduct sensitivity study about it.

→ Unfortunately, this was beyond the scope of this study; independent aerosol mass measurement was not available. However, with known chemical composition the mass distribution was converted to a number distribution using literature density values to yield a number value to conduct a CCN analysis. Although there is some error attributed to this method, it is consistently done from test to test.

L282 ' Combustion Aerosol, hereon also referred to as Black Carbon (BC)' Combustion aerosol is not equivalent as black carbon.

→ Yes, combustion aerosol is not all Black Carbon (BC).   We have clarified that BC is this paper is short for black carbon mixtures or BC containing particles and have revised the text in multiple places as follows:
→ Combustion aerosol or soot can form external and internal complex aerosol mixtures. Soot is considered insoluble but wettable (Lammel and Novakov 1995; Moore et al. 2014)  and the contributions of  Black Carbon containing particles to aerosol-cloud interactions at varied mixing states is not well known or understood (Lammel and Novakov 1995; Novakov and Corrigan 1996; Weingartner et al. 1997; Dusek et al. 2006; Kuwata and Kondo 2008; Koehler et al. 2009; Bond et al. 2013; McMeeking et al. 2011; Liu et al. 2013; Rojas et al. 2015). Thus, the ability of black carbon to mix with inorganic and organic compounds and to observe the extent of mixing as they activate as CCN is of great interest.   Prior to investigating the impact of mixing fresh combustion emissions with inorganic and organic aerosols, the CCN activation spectra of soot was measured using combustion aerosol generated from the APG.  The aerosol is likely composed of black carbon and oxidized carbonaceous material (Moore et al. 2014; Durdina et al. 2016). Thus we also refer to carbonaceous aerosol as black carbon mixtures (simply, BC mixtures).

L283 'BC is considered insoluble but wettable' Please show experimental data/cite literature to support the idea.

→ Previous work (e.g. but not limited to, Lammel and Novakov 1995; Moore et al. 2014) has indicated BC as hydrophobic, but still wettable.
→ It is understood that CCN can be soluble or insoluble but wettable particles - a water film can cover the surface of the aerosol and the activation of the particle is more dependent of the curvature and less of the composition. This is further supported by this study which showed a low activation diameter (fig. 7) for combustion aerosol and thus a low kappa value of approximately 0.001, which is close to 0 - particles that are insoluble but wettable have been characterized with a kapps value of close to 0.

L288 'The single sigmoid fit suggests that the aerosol generated is a homogenous mixture of black and brown carbon.' The activation curve is very broad. I do not think that such a broad activation curve could be observed if all the particles have exactly the same chemical composition. I wonder what the authors wanted to mean by 'homogeneous.' How can the authors make sure that brown carbon existed in the particles they measured? Please demonstrate the evidence for it.

→ A homogeneous mixture similar chemical compositional fractions across different particle sizes.  A homogenous mixture is an internal mixture.    The data presented for known compounds illustrated that homogenous (internal) mixtures exhibit singular curves.  That is the chemical composition across CCN active sizes is constant and thus the activation is smooth (with no plateau).  Externally mixed CCN exhibit plateaus and multiple points of activation. The CCN activation of the soot is a singular curve and thus indicates that the aerosol are internally (homogeneously) mixed.
→ We have removed the word brown carbon and replaced the wording to reflect the more oxidized soot components found in aerosol generated from flame burners.
→ The text is now rewritten as :There are no plateaus in the activation curve and the single sigmoid fit indicates that the aerosol generated is a homogenous internal mixture.

L292 'It is noted that the apparent hygroscopicity does not account for non-spherical

fractal particles.' I do not understand what this sentence means. Do the authors want to clarify that these particles do not have spherical shape?

→ The apparent hygroscopicity is calculated from the electrical mobility diameter and thus assumes all particles as spherical. Electrical mobility diameter measurement does not take into account non-spherical fractal particles.
→ The text has been changed as follows: It is noted that the apparent hygroscopicity is defined by the electrical mobility diameter that assumes particles are spherical.

[revised manuscript text omitted]

---

## Author Comment (AC2) · 10 May 2019

The manuscript represents a method of CCN data analysis for different mixtures of organic and inorganic components of varying mixing states with regards to the observed activation and hygroscopicity. By conducting controlled laboratory experiments of mixing, them transition from external to internal mixtures are effectively mimicked. For mixtures of known hygroscopicity CCN behavior agrees well with traditional Köhler theory. Finally, aerosol water is shown to play a significant role in promoting mixing and can be used to modify mixing states.
The paper is well written and easy to follow, though there are some issues and more thorough discussion should be made in specific sections. A very interesting point of the study is the deconvolution of the activation curves to different sigmoidal curves consistent with those of the different components, as well as the study of the impact of mixing fresh combustion emissions with inorganic and organic aerosols.

Specific comments:
1) More thorough discussion should be made in the study of mixing fresh combustion aerosol with inorganic and organic aerosols. Why is combustion aerosol considered and referred to solely as Black Carbon? Also, the activation curve presented in Fig.07 does not reach an obvious plateau, how can we speak about a "homogeneous mixture of black and brown carbon"? Also I would expect that as a homogenous mixture, the activation curve would be a lot steeper, especially in the larger particle size range. Can you please clarify/comment on this?

→ The combustion aerosol generated by the APG is used as a black carbon surrogate; it was not fully speciated in this study. The miniCAST utilized in the APG has been well characterized in previous work (e.g., Pinho et al. 2008; Seong and Boehman 2012; Mamakos et al. 2013; Maricq and Matti Maricq 2014; Durdina et al. 2016; Moore et al. 2014) . The combustion aerosol does contains black and oxidized carbon. Since there are no plateaus in the CCN activation of the soot, (it is a single smooth curve) this suggests that the size-resolved chemical composition is a homogenous (internal) mixture. The slope is indicative of activation at higher mobility diameters where it is known that the transfer function of the DMA is wider, thus contributing to a shallow slope (e.g., Lance et al., 2013).
→ We have revised the text with this added information.

2) A more thorough review of the current literature should be made, as there are relevant studies that are not mentioned. E.g. L69-70 Moore et al., 2012 provide an in-depth analysis of ambient CCN measurements during the 2010 Deepwater Horizon Oil Spill, both studying hygroscopicity vs mixing state and type of size distributions and droplet activation kinetics.

→ We have added the suggested references.

→ Moore et al, 2012 is one of many papers that studies changes of ambient CCN with mixing state. We have included the reference. An extensive literature review will show that there are many papers discuss aerosol mixing state and cloud condensation nuclei. We have added them in this revision. However, The majority of theses papers use ambient data sets of unspeciated aerosol composition. To our knowledge, we are the only manuscript to show changes in laboratory CCN with controlled changes in mixing state aerosol population. This is the novelty of the presented technique. Previous papers (dating back to the early 70's) have provided the theory and have shown the activation of laboratory externally mixed aerosol and internally mixed systems, separately. We have yet to find a paper that has controlled and measured the change from external to internal in the laboratory. We regret that this uniqueness did not come across in the first revision of the paper and thus we have spent considerable effort emphasizing the importance of the new technique presented in the revised manuscript.

→ The following text has been revised as follows.

Knowledge of the mixing state and the chemical composition can greatly improve CCN predictions and has been the focus of several studies (e.g., but not limited to (Bilde and Svenningsson 2004; Abbatt et al. 2005; Henning et al. 2005; Svenningsson et al. 2006; King et al. 2007; Cubison et al. 2008; Kuwata and Kondo 2008; Zaveri et al. 2010; Su et al. 2010; Wang et al. 2010; Spracklen et al. 2011; Ervens et al. 2010; Asa-Awuku et al. 2011; Liu et al. 2013; Jurányi et al. 2013; Paramonov et al. 2013; Padró et al. 2012; Moore et al. 2012; Meng et al. 2014; Bhattu and Tripathi 2015; Almeida et al. 2014; Schill et al. 2015; Crosbie et al. 2015; Che et al. 2016; Ching et al. 2016; Mallet et al. 2017; Sánchez Gácita et al. 2017; Cai et al. 2018; Schmale et al. 2018; Mahish et al. 2018; Kim et al. 2018; Chen et al. 2019; Stevens and Dastoor 2019)

3) CCN activity also depends on the atmospheric processing as well. L50-51 it is stated that CCN activity is complicated when inorganic salts are internally mixed with a complex organic. It has been shown in filed studies that even within a few hours of processing, hygroscopicity (thus activity as well) becomes more or less constant, e.g. Lathem et al., 2013; Bougiatioti et al., 2009. This, to my opinion, should be mentioned.

→ We agree CCN activity and the mixing state of the population depends on atmospheric processing. We have included this in the text. We have added the suggested additional references. Again, our challenge here is that there are a lot of global ambient CCN activity measurements that show changing CCN behaviour. The models to describe changes in CCN mixtures have been validated with ambient measurements. To our knowledge, we are the first to validate with known composition and control subsequent CCN from external to internal mixtures in the laboratory.

Technical corrections:

L109 What type of SMPS, sizing range and of CCNc? It is mentioned in Section 2.3. but it seems to me as missing information to me at that early stage.

→ The instrument specifics were organized into a single section under 'measurement and instrumentation'. The text has been updated with text directing the reader to refer to section 2.3 for instrument specifications

L293 What do the authors mean by "does not account for non-spherical fractal particles"?

→ The apparent hygroscopicity is calculated from the electrical mobility diameter that assumes particles are spherical. Electrical mobility diameter measurement does not take into account non-spherical fractal particles. We have elaborated this in revised text.

Fig.05 Some points exhibit huge variation
→ The variation can be attributed to the conversion from AMS mass to number.

**→ **References.**

1. Abbatt, J. P. D., Broekhuizen, K. and Pradeep Kumar, P.: Cloud condensation nucleus activity of internally mixed ammonium sulfate/organic acid aerosol particles, Atmos. Environ., 39(26), 4767–4778, 2005.
2. Asa-Awuku, A., Moore, R. H., Nenes, A., Bahreini, R., Holloway, J. S., Brock, C. A., Middlebrook, A. M., Ryerson, T. B., Jimenez, J. L., DeCarlo, P. F., Hecobian, A., Weber, R. J., Stickel, R., Tanner, D. J. and Huey, L. G.: Airborne cloud condensation nuclei measurements during the 2006 Texas Air Quality Study, J. Geophys. Res., 116(D11), doi:10.1029/2010JD014874, 2011.
3. Bhattu, D. and Tripathi, S. N.: CCN closure study: Effects of aerosol chemical composition and mixing state, J. Geophys. Res. D: Atmos., 120(2), 766–783, 2015.
4. Bilde, M. and Svenningsson, B.: CCN activation of slightly soluble organics: the importance of small amounts of inorganic salt and particle phase, Tellus B Chem. Phys. Meteorol., 56(2), 128–134, 2004.
5. Bond, T. C., Doherty, S. J., Fahey, D. W., Forster, P. M., Berntsen, T., DeAngelo, B. J., Flanner, M. G., Ghan, S., Kärcher, B., Koch, D., Kinne, S., Kondo, Y., Quinn, P. K., Sarofim, M. C., Schultz, M. G., Schulz, M., Venkataraman, C., Zhang, H., Zhang, S., Bellouin, N., Guttikunda, S. K., Hopke, P. K., Jacobson, M. Z., Kaiser, J. W., Klimont, Z., Lohmann, U., Schwarz, J. P., Shindell, D., Storelvmo, T., Warren, S. G. and Zender, C. S.: Bounding the role of black carbon in the climate system: A scientific assessment, J. Geophys. Res. D: Atmos., 118(11), 5380–5552, 2013.
6. Cai, M., Tan, H., Chan, C. K., Qin, Y., Xu, H., Li, F., Schurman, M. I., Liu, L. and Zhao, J.: The size-resolved cloud condensation nuclei (CCN) activity and its prediction based on aerosol hygroscopicity and composition in the Pearl Delta River (PRD) region during wintertime 2014, Atmos. Chem. Phys., 18(22), 16419–16437, 2018.
7. Chen, L., Li, Q., Wu, D., Sun, H., Wei, Y., Ding, X., Chen, H., Cheng, T. and Chen, J.: Size distribution and chemical composition of primary particles emitted during open biomass burning processes: Impacts on cloud condensation nuclei activation, Sci. Total Environ., 674, 179–188, 2019.
8. Ching, J., Riemer, N. and West, M.: Black carbon mixing state impacts on cloud microphysical properties: Effects of aerosol plume and environmental conditions, JOURNAL OF GEOPHYSICAL RESEARCH-ATMOSPHERES, 121(10), 5990–6013, 2016.
9. Crosbie, E., Youn, J.-S., Balch, B., Wonaschütz, A., Shingler, T., Wang, Z., Conant, W. C., Betterton, E. A. and Sorooshian, A.: On the competition among aerosol number, size and composition in predicting CCN variability: a multi-annual field study in an urbanized desert, Atmos. Chem. Phys., 15, 6943–6958, 2015.
10. Cubison, M. J., Ervens, B., Feingold, G., Docherty, K. S., Ulbrich, I. M., Shields, L., Prather, K., Hering, S. and Jimenez, J. L.: The influence of chemical composition and mixing state of Los Angeles urban aerosol on CCN number and cloud properties, Atmos. Chem. Phys., 8(18), 5649–5667, 2008.
11. Durdina, L., Lobo, P., Trueblood, M. B., Black, E. A., Achterberg, S., Hagen, D. E., Brem, B. T. and Wang, J.: Response of real-time black carbon mass instruments to mini-CAST soot, Aerosol Science and Technology, 50(9), 906–918, doi:10.1080/02786826.2016.1204423, 2016.
12. Dusek, U., Reischl, G. P. and Hitzenberger, R.: CCN activation of pure and coated carbon black particles, Environ. Sci. Technol., 40(4), 1223–1230, 2006.
13. Ervens, B., Cubison, M. J., Andrews, E., Feingold, G., Ogren, J. A., Jimenez, J. L., Quinn, P. K., Bates, T. S., Wang, J., Zhang, Q., Coe, H., Flynn, M. and Allan, J. D.: CCN predictions using simplified assumptions of organic aerosol composition and mixing state: a synthesis from six different locations, Atmos. Chem. Phys., 10(10), 4795–4807, 2010.
14. Henning, S., Rosenørn, T., D'Anna, B., Gola, A. A., Svenningsson, B. and Bilde, M.: Cloud droplet activation and surface tension of mixtures of slightly soluble organics and inorganic salt, Atmos. Chem.

Phys., 5(2), 575–582, 2005.

[revised manuscript text omitted]

---

## Referee Report (RR1)

The manuscript "External and Internal Cloud Condensation Nuclei (CCN) Mixtures: Controlled Laboratory Studies of Varying Mixing States" presents the results of testing and validating the CCN activation with multicomponent and varying mixing state under controlled laboratory conditions. Mixing state is important for CCN concentrations and it is interesting to test the relationship the CCN activation with mixing state in laboratory studies.

I did not read the manuscript of previous version. For this version, I have a few specific comments.

1. The manuscript claims that "the aerosol mixing state can be observed in CCN activation data and can thus be revisited in complex aerosol data sets to understand the extent of mixing" (L392-394, and something similar in the abstract (L39-40)). In this study, two-components mixture were used and a plateau was found for the external mixing case. I am curious how well the plateau can be resolved when aerosol contains more than two components, say three or five components. It might look like a continuous activation curve. Therefore, it would be helpful to discuss the limit of this approach.

   Also L286-288, "Results utilizing CCN activation data for aerosols <50nm may be a good substitute for estimating aerosol chemical fractions when other instruments with lower size resolution are not readily available." This may be difficult if aerosol contains more than two components.

2. Fig. 3, in the caption, it is mentioned that "Dashed lines indicate 20% uncertainty." What does this uncertainty exactly mean? The lines are of 0.8:1 and 1.2:1 lines? It seems to be not the case according the values in the figure. I have similar questions for Fig. 5 and Fig. 6.

3. L261-264, it is very interesting to attribute the change of mixing state to minute water content. Have the authors also considered the possibility of effect of heating on the size of SA particles? If the size distribution changes, the coagulation efficiency of SA and AS may change, which will also affect the mixing state.

4. L332-334, it would be helpful for readers to understand if the $D_{50}$ values of NaCl and AS were provided.

5. Legends of Fig. 9 are missing.

6. L380-382, "…which have attributed increases in plateau height to ***the extent of internal mixing of hygroscopic materials on externally mixed inactivated aerosols.***" This phrasing seems to be confusing for me.

---

## Author Response (AR2)

⇨ We thank the Associate Editor for comments and the opportunity to revise the manuscript. We have taken care to address each comment in a point by point response.

Responses to Report #3

The manuscript "External and Internal Cloud Condensation Nuclei (CCN) Mixtures: Controlled Laboratory Studies of Varying Mixing States" presents the results of testing and validating the CCN activation with multicomponent and varying mixing state under controlled laboratory conditions. Mixing state is important for CCN concentrations and it is interesting to test the relationship the CCN activation with mixing state in laboratory studies.

I did not read the manuscript of previous version. For this version, I have a few specific comments.

1. The manuscript claims that "the aerosol mixing state can be observed in CCN activation data and can thus be revisited in complex aerosol data sets to understand the extent of mixing" (L392-394, and something similar in the abstract (L39-40)). In this study, two-components mixture were used and a plateau was found for the external mixing case. I am curious how well the plateau can be resolved when aerosol contains more than two components, say three or five components. It might look like a continuous activation curve. Therefore, it would be helpful to discuss the limit of this approach.

   Also L286-288, "Results utilizing CCN activation data for aerosols <50nm may be a good substitute for estimating aerosol chemical fractions when other instruments with lower size resolution are not readily available." This may be difficult if aerosol contains more than two components.

   ⇨ Discussion about the limit of multiple components has been added to the text as follows:
   ⇨ L251. For more than 2-component externally mixed particle distributions, more than two plateaus will be observed. For example, the work of Schill et al. 2015 shows multiple plateaus for a five-component external mixture mimic of ocean spray CCN. If one considers a limiting case of infinite components with distinct varying degrees of hygroscopicity, then the activation curve will be monotonic below 1 and may appear to be representative of an internal mixture; the CCN activation curve will approach the shape of a shallow sigmoidal slope (however not as steep or instantaneous as the ideal step function)
   ⇨ L294"Results with distinct CCN activation plateaus, especially of 2-3 distinct hygroscopicities, may be useful for estimating aerosol chemical fractions when other instruments with lower size resolution are not readily available"

2. Fig. 3, in the caption, it is mentioned that "Dashed lines indicate 20% uncertainty." What does this uncertainty exactly mean? The lines are of 0.8:1 and 1.2:1 lines? It seems to be not the case according the values in the figure. I have similar questions for Fig. 5 and Fig. 6.

⇨ That is an error in the text. The dashed lines in Figures 3,5, and 6 represent 50% uncertainty from the 1:1. The lines are 1.5:1 and 1:1.5 and were kept consistent across the figures.  The data in Figure 3 are well within the 50% uncertainty, and specifically within 20%  uncertainty from the agreement with theory.  The error bars are the standard deviation from replicate measurements. We have added this additional clarification to the revised manuscript and have made sure that the text is congruent with the figures.

⇨ L228 "Regardless of surface tension omissions, the single sigmoidal experimental data set is within 20% of theoretical agreement and indicative of a single component or homogeneous internally mixed aerosol population."

3. L261-264, it is very interesting to attribute the change of mixing state to minute water content. Have the authors also considered the possibility of effect of heating on the size of SA particles? If the size distribution changes, the coagulation efficiency of SA and AS may change, which will also affect the mixing state.

⇨ The effects of heating on semi-volatile succinic acid particles is interesting but was not discussed because we did not observe evidence of potential evaporation affecting our SA and AS results. Additionally, single-particle composition measurements at low temperatures are required to robustly quantify the effects of coagulation on mixing state. We unfortunately did not have access to such instrumentation. We expand this explanation with the points below.

⇨ For example,  In Figure 4, the effects on the CCN activation curve of SA and AS are presented. When heating is on, distinct plateaus (below 1) are observed indicating that SA is present and contributes to a fraction of the aerosol at those smaller sizes. It should be emphasized that the heating of concern occurs during aerosol generation.  The subsequent aerosol that remains is then characterized for CCN activity. The plateau below one is an indication of multiple components at the given sizes.  Additionally, kappa is a particle intrinsic property;  that is regardless of particle size distribution, kappa hygroscopicity is a reflection of the chemical composition of particles. Simply stated, ten 100nm particles will have the same hygroscopicity as one thousand 50 nm particles. Even if the the distributions change (if the 50 nm particles evaporate), the hygroscopicity of the component should stay the same.  When two sigmoids are applied in our data sets, we correctly measure the kappa of the two distinct components. If all the SA was evaporated, then the critical activation diameters and kappa values would look entirely like AS.

⇨ It is noted that particle evaporation has the potential to modify the volume fraction of a particle at a given size. That is for external mixtures, the plateau height will likely be higher for the more hygroscopic and less volatile material.  Figure 6 shows that the AMS measured fraction of potentially evaporated SA/AS mixtures agrees reasonably well with the fraction estimated from the CCN at different supersaturations.    For an internal mixture, the asymptote equals 1 and  if the size resolved contribution of each component is known, there is a shift in the CCN activation from left to right (as is observed in Figure

3). Our estimates of the volume fraction from kohler theory agree with the known quantities used to make the solution. These particles were also heated during particle activation. Again, we see little evidence of the effects of succinic acid evaporation in our data set. In the transition mixing state experiments from external to internal, we do not have single-particle information and thus cannot fully quantify the volume fraction at each size. The reported CCN activation curve and resulting kappa values during the transition from external to internal also agree with theory. That is the external mixture agrees with AS and SA theoretical values and the internal mixtures is somewhere between the two, again suggesting that SA is present and has not completely evaporated during the measurement. We have provided some additional discussion to address concerns about volatile species and the coagulation of particles.

⇨ **Additional revised text**
⇨ L253. "It is noted that heated succinic acid particles can evaporate during aerosol generation and before CCN measurement; the asymptote in Figure 4 is an indication that multiple components are present in the total aerosol distribution.
⇨ L218. The shifting of the CCN activation sigmoid (left and right) is also expected of internally mixed particles formed via the coagulation of separate particle distributions (Farmer et al., 2015).

.

4. L332-334, it would be helpful for readers to understand if the D50 values of NaCl and AS were provided.
⇨ The additional information has been provided in the revised text.
⇨ L343 "Thus for both the AS and NaCl external mixtures, the activation diameters derived from a singular fit were consistent with the expected dp50 < 50 nm of the respective inorganic salts. Specifically at Sc=1.1%, AS and NaCl particles activated at  25 and 19 nm  respectively (congruent with theoretical dp50 at 24.8 and 19.0 nm)"

5. Legends of Fig. 9 are missing.
⇨ The following text has been revised.
⇨ "Figure 9. a) Time series of CCN/CN activated fractions of Succinic Acid (SA) and combustion aerosol (BC) mixture in flow tube. b) The CCN/CN activated fraction (closed triangles) of Succinic Acid and Combustion Aerosol mixtures for particle distribution scans 24, 45, 54, 73, and 94. Aerosol water is introduced at ~ scan 70 to promote internal mixing. Cross symbols show the particle  size distribution mixed aerosol."

6. L380-382, "...which have attributed increases in plateau height to the extent of internal mixing of hygroscopic materials on externally mixed inactivated aerosols." This phrasing seems to be confusing for me.
⇨ The text has been revised and now reads.
⇨ "This is consistent with CCN spectra observed in ambient studies, which have attributed observations  in CCN activation plateau heights less than one to the contributions of externally mixed and inactivated (typically non-hygrosocopic black carbon) aerosols".

Vu et al. claim that 'very' dry SA particles remain externally mixed with other aerosol particles until a preheating step is removed resulting in dry SA particles – and then homogeneous internally mixed particles occur in their flow tube. The authors do not present a reasonable explanation for such unexpected behavior. However, it is clear that this interpretation cannot be justified from the data presented in Fig. 9. As long as the pre-heating step is applied, the SA particles seem to be dominated by a mode centered near a mobility diameter of ~20 nm (e.g. scan 54). When the preheating is turned off, then the resulting particle number size distribution has a clear maximum near 80-85 nm decreasing significantly towards 130 nm (e.g. scan 73), where the soot particle size distribution supposedly has a maximum. A reasonable explanation for the behavior is a transition from very pronounced evaporation of SA particles, to a somewhat lower evaporation of SA particles, which leads to the observed transition in the CCN spectra and dominance of SA in the size range studied.

Hence, the interpretation of the behavior of the experimental system presented does not seem appropriate. It would be meaningful to characterize the system with more well-behaved substances and also with additional instrumentation more sensitive to the mixing state than a CCNC setup. Furthermore, the system should be characterized with more substances, if indeed the system behavior is very sensitive to the aerosol RH, as claimed by the authors.

The issues described above also influence the interpretation and use of results presented for internally mixed SA-NaCl particles as well as the externally mixed SA-ammonium sulfate particles.

⇨ The effects of heating on semi-volatile succinic acid particles is interesting but was not discussed because we did not observe evidence of potential evaporation affecting our SA and AS results. We expand this explanation with the points below.

⇨ For example, In Figure 4, the effects on the CCN activation curve of SA and AS are presented. When heating is on, distinct plateaus (below 1) are observed indicating that SA is present and contributes to a fraction of the aerosol at those smaller sizes. It should be emphasized that the heating of concern occurs during aerosol generation. The subsequent aerosol that remains is then characterized for CCN activity. The plateau below one is an indication of multiple components at the given sizes. Additionally, kappa is a particle intrinsic property; that is regardless of particle size distribution, kappa hygroscopicity is a reflection of the chemical composition of particles. Simply stated, ten 100nm particles will have the same hygroscopicity as one thousand 50 nm particles. Even if the distributions change (if the 50 nm particles evaporate), the hygroscopicity of the component should stay the same. When two sigmoids are applied in our data sets, we correctly measure the kappa of the two distinct components. If all the SA was evaporated, then the critical activation diameters and kappa values would look entirely like AS.

⇨ It is noted that particle evaporation has the potential to modify the volume fraction of a particle at a given size. That is for external mixtures, the plateau height will likely be higher for the more hygroscopic and less volatile material. Figure 6 shows that the AMS measured fraction of potentially evaporated SA/AS mixtures agrees reasonably well with the fraction estimated from the CCN at different supersaturations. For an internal mixture, the asymptote equals 1 and if the size resolved contribution of each component is known, there is a shift in the CCN activation from left to right (as is observed in Figure 3). Our estimates of the volume fraction from kohler theory agree with the known quantities used to make the solution. These particles were also heated during particle activation. Again, we see little evidence of the effects of succinic acid evaporation in our data set. In the transition mixing state experiments from external to internal, we do not have single-particle information and thus cannot fully quantify the volume fraction at each size. The reported CCN activation curve and resulting kappa values during the transition from external to internal also agree with theory. That is the external mixture agrees with AS and SA theoretical values and the internal mixtures is somewhere between the two, again suggesting that SA is present and has not completely evaporated during the measurement. We have provided some additional discussion to address concerns about volatile species and the coagulation of particles.

⇨ In Figure 9, the changes in activation curve are consistent with the transition from external to internally mixed aerosol. SA (whether partially evaporated) is present and observed in the data set. The change in size distributions of SA does not change its inherent hygroscopicity. The concern with evaporation should be most relevant for the externally mixed case but as pointed out by the reviewer there are particles at 20nm present. Once heating is turned off, more SA could be present. However, single particle measurements are required to quantify the particle volumes at activation for the transition to internally mixed aerosols. We do not have those measurement capabilities. The utility of the work here shows that CCN data may be used to observe the mixing state and can be used to estimate theoretical contributions in the future.